# Identification of Immuno-Targeted Combination Therapies Using Explanatory Subgroup Discovery for Cancer Patients with EGFR Wild-Type Gene

**DOI:** 10.3390/cancers14194759

**Published:** 2022-09-29

**Authors:** Olha Kholod, William Basket, Danlu Liu, Jonathan Mitchem, Jussuf Kaifi, Laura Dooley, Chi-Ren Shyu

**Affiliations:** 1MU Institute for Data Science and Informatics, University of Missouri, Columbia, MO 65212, USA; 2Department of Electrical Engineering & Computer Science, University of Missouri, Columbia, MO 65212, USA; 3Department of Surgery, School of Medicine, University of Missouri, Columbia, MO 65212, USA; 4Harry S. Truman Memorial Veterans’ Hospital, Columbia, MO 65201, USA; 5Department of Otolaryngology, School of Medicine, University of Missouri, Columbia, MO 65212, USA

**Keywords:** immuno-targeted combination therapies, subgroup discovery, cancer

## Abstract

**Simple Summary:**

Cancer immunotherapy is a form of cancer treatment that uses a person’s own immune system to prevent, control, and eliminate cancer. However, immunotherapy alone may not be effective, especially in patients with limited treatment options. Immuno-targeted combination therapies have a potential to create synergetic effects with improved health outcomes. Therefore, there is a growing interest in searching for therapeutic combinations that could extend the benefits of immunotherapy. In this study, we designed a computational method that facilitated the identification of effective combination therapies for cancer patients with few treatment options. We determined several specific drug targets that substantially increased the odds of stable disease versus progressive disease for head and neck cancer, lung cancer, and melanoma. The identified treatment combinations were targets in several clinical trials. Moreover, our approach has the potential to improve the selection of patients for immuno-targeted combination therapies and lead to an overall improvement in health outcomes for cancer patients with limited treatment options.

**Abstract:**

(1) Background: Phenotypic and genotypic heterogeneity are characteristic features of cancer patients. To tackle patients’ heterogeneity, immune checkpoint inhibitors (ICIs) represent some the most promising therapeutic approaches. However, approximately 50% of cancer patients that are eligible for treatment with ICIs do not respond well, especially patients with no targetable mutations. Over the years, multiple patient stratification techniques have been developed to identify homogenous patient subgroups, although matching a patient subgroup to a treatment option that can improve patients’ health outcomes remains a challenging task. (2) Methods: We extended our Subgroup Discovery algorithm to identify patient subpopulations that could potentially benefit from immuno-targeted combination therapies in four cancer types: head and neck squamous carcinoma (HNSC), lung adenocarcinoma (LUAD), lung squamous carcinoma (LUSC), and skin cutaneous melanoma (SKCM). We employed the proportional odds model to identify significant drug targets and the corresponding compounds that increased the likelihood of stable disease versus progressive disease in cancer patients with the EGFR wild-type (WT) gene. (3) Results: Our pipeline identified six significant drug targets and thirteen specific compounds for cancer patients with the EGFR WT gene. Three out of six drug targets—FCGR2B, IGF1R, and KIT—substantially increased the odds of having stable disease versus progressive disease. Progression-free survival (PFS) of more than 6 months was a common feature among the investigated subgroups. (4) Conclusions: Our approach could help to better select responders for immuno-targeted combination therapies and improve health outcomes for cancer patients with no targetable mutations.

## 1. Introduction

Immunotherapy represents one of the most promising therapeutic approaches in cancer [1,2]. Immune checkpoint inhibitors (ICIs) can lead to long-term remission and improved survival in patients with locally advanced/metastatic cancer [3,4]. However, almost 50% of cancer patients that are eligible for treatment with ICIs do not respond [5,6]. This is particularly true in patients without targetable mutations [7]. Conventional cancer treatments, such as radiation therapy [8], cytotoxic chemotherapy [9,10], and targeted therapy [11,12], have immunomodulatory effects along with direct tumor-cell-killing activities. Their clinical utility in combination with ICIs potentially creates synergetic effects with improved and durable clinical response [13,14]. Therefore, there is a growing interest in searching for predictive biomarkers of therapeutic response and identifying therapeutic targets that could extend the benefits of ICIs.

Recent studies have shown that various factors contribute to the response to ICIs [15]. The tumor mutational burden (TMB) has emerged as one of the most crucial factors to determine the efficacy of ICIs [16,17]. The TMB quantitatively assesses the number of mutations per megabase (muts/Mb). High TMB values are an indication of better survival in cancer patients receiving immunotherapy [18]. However, the effectiveness of immunotherapy has been demonstrated in some patients with a low TMB, while unfavorable outcomes have been observed in a significant number of patients with a high TMB [19,20]. It is also unclear how patients that harbor specific genetic alterations would respond to immunotherapy. For example, anti-PD-1 monotherapy has been shown to be unable to improve survival outcomes for non-small cell lung cancer (NSCLC) patients with EGFR mutations, even in patients with high PD-L1 expression [21]. Meanwhile, the ATLANTIC trial has disputed this observation and has highlighted the benefits of ICIs for EGFR-mutated tumors [22]. Therefore, the question of how to effectively utilize thousands of cell-intrinsic and -extrinsic characteristics to identify patient subpopulations that would benefit from combinatorial treatments with ICIs remains unanswered.

Over the past decades, Subgroup Discovery (SD) methods have been used to find homogenous subpopulations of patients that share common genetic profiles and may respond similarly to therapeutic regimens [23,24]. Existing SD approaches can be classified into two major categories: statistical methods and data-mining methods. Statistical methods include regression analysis [25], clustering techniques [26], and latent class analysis (LCA) [27]. For example, LCA is a finite mixture model that aims to uncover unobserved groups within a population. However, this technique does not automatically determine the number of latent classes and produces solutions that heavily rely upon expert knowledge, which significantly limits the capability to discover novel subgroups in large heterogenous cancer datasets. Data-mining methods for SD comprise two major categories based on the search strategy for potential candidates—heuristic approaches and exhaustive approaches [28]. For example, CN2-SD [29] is a heuristic algorithm that searches for the statistically “most interesting” subgroups that are as large as possible and have the most unusual distributional characteristics with respect to the property of interest. However, CN2-SD suffers from the standard scaling problem that appears in the evaluation of large datasets, including heterogenous cancer datasets.

In this work, we extended our Subgroup Discovery algorithm [30] to predict immuno-targeted combination therapies for EGFR WT patient subpopulations. We focused on EGFR WT subgroups because EGFR-TKIs are widely used for the first-line treatment of patients with EGFR-sensitizing mutations, leading to longer progression-free survival (PFS) [31,32]. However, beyond the first line, especially for EGFR WT tumors, the role of EGFR-TKIs is elusive. We employed a proportional odds model to identify significant drug targets and the corresponding compounds that increased the likelihood of stable disease versus progressive disease in cancer patients that had the EGFR WT gene. This approach could help to better select responders for immuno-targeted combination therapies and improve health outcomes for cancer patients with no targetable mutations.

## 2. Materials and Methods

Our informatic framework consists of two modules: Subgroup Discovery and Immuno-Targeted Combination Therapies Discovery. The Subgroup Discovery module identifies homogenous patient subgroups based on both phenotypic and genotypic parameters and explains the differences among these subgroups using gene expression patterns. The Immuno-Targeted Combination Therapies Discovery module predicts potential drug targets and the corresponding compounds for uses in combination therapies with immunotherapy for cancer patients with no targetable mutations. The details of the modules are described below.

### 2.1. Data Mapping

The TCGA dataset from Pan Cancer Atlas [33] consists of 1952 cancer patients. We focused on four different cancer types: (1) head and neck squamous carcinoma (HNSC), *n* = 515; (2) lung adenocarcinoma (LUAD), *n* = 510; (3) lung squamous carcinoma (LUSC), *n* = 484; and (4) skin cutaneous melanoma (SKCM), *n* = 443. We chose these cancer types because in several clinical trials involving patients with advanced lung cancer, skin cutaneous melanoma, and other solid tumors, they demonstrated significant response rates to nivolumab and pembrolizumab, where both are monoclonal IgG4 antibodies against PD-1 [34].

The input data consisted of phenotypic and genotypic variables for a disease population. The phenotypic variables divided disease population into subgroups, while the genotypic variables described the main characteristics (patterns) of these subgroups. Eleven phenotypic variables were chosen for this study, including demographic data (e.g., Age, Gender), clinical–pathologic data (e.g., Tumor Type, Tumor Mutational Burden (TMB)), treatment history (e.g., Administration of Radiotherapy), and behavioral data (e.g., Smoking Status). As a part of the human-in-the-loop process, a physician panel specialized in the care of cancer patients selected the phenotypic variables to be included in the analyses. Many of these phenotypic variables were categorical. For example, Smoking Status was categorized as follows: (1) Never Smoker, (2) Current Smoker, and (3) Former Smoker. We ensured all continuous variables in the dataset were categorized based on the clinical literature or experience. To deal with missing data, we excluded patients with too many missing values in both phenotypic and genotypic variables and used “NA” as a new category to represent missing values. However, “NA” variables were never used to form subgroups.

The genotypic data in this study were z-score-transformed values for 730 immune-related genes (including PD-1 and PD-L1 genes) [35] and 40 housekeeping genes between normal and tumor tissues. The categorization of the genotypic variables was based on z-score-transformed values, where downregulated genes corresponded to the range (−Inf, −2); not significantly expressed genes corresponded to the range (−2, 2); and upregulated genes corresponded to the range (2, Inf). Therefore, each gene represented a genotypic variable.

### 2.2. Subgroup Discovery

#### 2.2.1. Patient Stratification

The main goal of the Subgroup Discovery module is to identify homogeneous patient subpopulations. A set of common phenotypic and genotypic parameters specifies each unique subgroup. For example, Females with a TMB defined as high (more than 10 muts/Mb) with SKCM could be considered as a subgroup. The unique subgroup is reevaluated each time a phenotypic variable is included. The statistical significance of each subgroup is defined using the genotypic patterns that distinguish this subgroup from the rest of the outer population.

The Subgroup Discovery module includes three levels: *Path Expansion*, *Floating Subgroup Selection*, and *Inclusion*/*Exclusion* criteria. This method differs from the decision tree algorithm in a way that allows the same patient to be a member of multiple unique subgroups. For example, LUSC patients could be members of the (LUSC, Former Smoker) subgroup and the (LUSC, Stage IIIA) subgroup. The key objective of the subgroup stratification process is to determine a large number of existing subgroups based on phenotypic parameters, where most patients in that subgroup shared unique genotypic patterns. This method is not greedy, because the algorithm tracks the top potential subgroups based on local optimal selection [36].

By iterating over various paths in the search space, the *Floating Subgroup Selection* traverses multiple paths. A phenotypic variable with the highest contrast score against the outer population (e.g., *Tumor Type* = *LUAD* in Figure 1) forms a base subgroup via the *Path Expansion* process. In the first inclusion step, it adds a new variable, e.g., *EGFR* = *WT*, to the base subgroup (*Tumor Type* = *LUAD* AND *EGFR* = *WT*). To eliminate less significant inclusion steps, the exclusion function is used after each inclusion step. For example, when in the third inclusion step, the subgroup is (*Tumor Type* = *LUAD* AND *EGFR* = *WT* AND *Smoker = Former* AND *Gender* = *Female*), the exclusion function eliminates the less significant move (*Smoker = Former*) from the existing subgroup if the newly created subgroup (*Tumor Type* = *LUAD* AND *EGFR* = *WT* AND *Gender* = *Female*) has a higher contrast score. When the algorithm reaches the subgroup with the highest contrast score, the exploratory search is terminated.

#### 2.2.2. Subgroup Contrast

The algorithm identifies the genotypic patterns that frequently occur within each candidate subgroup but are rare in the rest of the population. To evaluate the frequency for a specific genotypic pattern in the homogenous subgroup, the support metric [37] (Equation (1)) is utilized. The growth rate metric [38] (Equation (2)) is employed to evaluate the contrast of the pattern in the selected subgroup.
(1)$$Support(p,\;D)=\frac{\left|\langle D,\;p\rangle \right|}{\left|D\right|}$$
where *p* is a pattern, |<*D*, *p*>| is the number of patients with specific genotypic pattern, and |*D*| is the total number of patients in the collection.
(2)$$Growth(cp,\;{S_{G}}_{1},\;{S_{G}}_{2})=\frac{\mathrm{Max}\left|s_{1}{,\;s}_{2}\right|}{\mathrm{Min}\left|s_{1}{,\;s}_{2}\right|}$$
where *cp* is the contrast pattern, *S_G_*_1_ is the focus subgroup, *S_G_*_2_ is the outer population, *s*_1_ is the support of *cp* in the focus subgroup, and *s*_2_ is the support of *cp* in the outer population.

We employ a *J*-value [39] (Equation (3)) to calculate the contrast for all patterns in each subgroup.
(3)$$J\text{-}\mathrm{value}=\frac{{T\;\times \;J}_{org}{+M\;\times \;\bar{J}}_{avr}\;}{T+M}$$
where *T* is a parameter related to the population size preference that depends on the type of the disease under study and whether it is a rare disease or not, based on the concept of the Bayesian average [40].

There are six major parameters for the Subgroup Discovery module: *min_support_proportion*, *max_depth*, *max_breadth*, *min_improvement_significance*, *max_checks* and *max_pop_complexity*. The *min_support_proportion* parameter describes the minimum proportion of the subgroup rows that a pattern must be present in and was set to 0.05. The *max_depth* parameter represents the maximum pattern length and was adjusted to five. The *max_breadth* stands for the maximum number of candidates at each search level and was modified to 50. The *min_improvement_significance* reflects the *p*-value needed to add any new element to a pattern and was set to 0.05. The *max_checks* parameter describes a hard cap on the max number of patterns checked and was adjusted to 1000. Finally, the *max_pop_complexity* describes the maximum number of phenotypic variables that can define a subgroup and was modified to three.

### 2.3. Immuno-Targeted Combination Therapies Discovery

Our goal was to find significant drug targets and their corresponding compounds that could be used in immuno-targeted combination therapies for cancer patients with no targetable mutations. Specifically, we were interested in identifying significant drug targets that increased the likelihood of stable disease versus progressive disease in four cancer types: HNSC, LUAD, LUSC, and SKCM. For this purpose, we used the proportional odds model [41] on Prat, A., et al.’s dataset [42]. The dataset consists of 65 cancer patients, and 15 phenotypic and 770 genotypic variables. The outcome variable was categorical and had four levels: partial response, complete response, progressive disease, and stable disease. The continuous covariates were represented as genes with corresponding normalized gene expression values. We used the MASS R package [43] to fit the proportional odds model with the logit link function. We computed *p*-values via two-tailed z-tests to identify significant predictors of response to immunotherapy.

## 3. Results

### 3.1. Identification of Candidate Subgroups for Immuno-Targeted Combination Therapies

The overview of our informatic pipeline is presented in Figure 2. The Subgroup Discovery module generated 9887 subgroups. To reduce the search space, we focused on subgroups that had Tumor Type (HNSC, LUAD, LUSC, or SKCM) as one of the phenotypic variables. The filtering procedure resulted in 1207 subgroups. To further focus on more specific subgroups, we selected those that had at least three phenotypic variables. In total, 1129 subgroups were used in our computational experiment.

First, we investigated subgroup coverage in the whole dataset of 1948 cancer patients. The subgroup coverage ranged from 1.23% to 25.97%. Therefore, we decided to target subgroups or unions of subgroups that covered at least 20% of the whole dataset, which resulted in eleven subgroup unions. Secondly, we narrowed down our search to four subgroup unions that had both Tumor Type (HNSC, LUAD, LUSC, or SKCM) and the EGFR WT gene. We hypothesized that cancer patients from these four subgroup unions were unlikely to produce a favorable outcome upon treatment with EGFR inhibitors due to the lack of targetable mutations in this gene. However, these patients may have other mutations that can be sensitive to FDA-approved targeted treatments. Therefore, it is important to identify these additional drug targets to improve patients’ outcomes. In addition, immunotherapy alone may not produce durable responses for these patients. In the ongoing phase III KEYNOTE-042 trial of patients with treatment-naïve, advanced, EGFR/ALK WT NSCLC and at least 1% tumor PD-L1 expression, there was no statistically significant PFS benefit among patients receiving *Pembrolizumab* compared with those receiving chemotherapy, except for those with the highest level of PD-L1 expression [44]. However, EGFR WT cancer patients may benefit from compounds that are used for immuno-targeted combination therapies, as was previously showed in EGFR WT NSCLC patients [45]. Thirdly, we identified common differentially expressed (DE) genes for these four subgroup unions. We started with identifying unique genotypic patterns for each subgroup union and then determined unique elements from these genotypic patterns. The intersection of unique genes from four subgroup unions produced 380 DE genes. The summary for each subgroup union is presented in Table 1.

There were also some unique features for each subgroup union. Our analysis revealed that HNSC EGFR WT subgroups consisted of patients with a low TMB. The role of the TMB in predicting the outcome of immunotherapy for advanced HNSCC remains unclear. For example, one study showed that immunotherapy was more effective in metastatic HNSCC patients with a high TMB, and the median OS of these patients was 2.5 times as long as that of patients with a low TMB (25 vs. 10 months) [46]. However, data from over 10,000 patient tumors included in The Cancer Genome Atlas failed to support the application of a high TMB as a biomarker for treatment with ICIs in all solid cancer types [47]. The LUAD EGFR WT subgroups had a substantial number of patients with the BRAF WT gene. The clinical impact of BRAF mutational subtypes on lung adenocarcinoma remains to be defined. For example, the data from two large German lung cancer centers showed that patients with BRAF-mutated NSCLC had an inferior prognosis, which was not determined by the BRAF mutation functional class. However, in contrast to NSCLC with other driver mutations, BRAF-mutated NSCLC exhibited high susceptibility to immune checkpoint inhibitors [48]. The LUSC union consisted of patients who were former smokers. For example, LUSC patients with a history of smoking and a high TMB had longer PFS after treatment with chemoimmunotherapy or anti-angiogenesis therapy [49]. There were no distinctive phenotypic features for the SKCM union. These identified unique features should be considered when designing combinatorial treatment regimens for EGFR WT cancer patients.

### 3.2. Drug Target Prediction for EGFR Wild-Type Subgroups

We examined the Drug Gene Interaction Database (DGIdb) [50] to map therapeutic compounds that could be used in immuno-targeted combination therapies. This search produced 155 potential drug targets and 36 targeted treatments. We then mapped the 380 common DE genes from the subgroup unions to the list of 155 drug targets. In this analysis, we identified 25 targets and 16 compounds that may be used in combination with ICIs.

The Immuno-Targeted Combination Therapies Discovery module identified six significant drug targets: CDH5, FCGR2B, IGF1R, ITK, JAK2, and KIT (Appendix A). The estimates in the output were given in units of ordered logits or ordered log odds. For example, for the FCGR2B gene, the likelihood of stable disease versus partial response, complete response, and progressive disease increased by 9.70 on the log odds scale. We also calculated the odds ratios and confidence intervals (Appendix A). These were obtained either by profiling the likelihood function or using the standard errors and assuming a normal distribution (Table 2). For example, for every unit increase in the expression of the KIT gene, the odds of having stable disease was multiplied by 3.57 times, with all other variables remaining constant.

Therefore, these six drug targets were predicted to benefit EGFR WT patients in immuno-targeted combination therapies (Table 3).

To address the question about the potential drug interaction effects and toxicity of identified combinations, we employed the Drug Interaction Checker from the Drug Bank database [51]. Based on our search, the administration of PD-1 or PD-L1 inhibitors in combination with FCGR2B-targeting therapies could increase the risk of adverse effects. Specifically, these therapeutic combinations carry a risk of immunogenicity, which can produce a wide array of adverse effects, the most serious of which include anaphylaxis and serum sickness-type reactions [52]. A few studies suggested that the use of multiple immunoglobulin agents is relatively safe and may be more effective than monotherapy under certain conditions [53,54].

From clinicaltrials.gov, we were able to identify clinical trials for predicted immuno-targeted combination treatments (Table 4).

Some of the clinical trials from Table 4 are still ongoing; therefore, they do not have information about primary and secondary outcomes. However, we were able to retrieve such information from other clinical trials, such as NCT02039674 and NCT01472081. In the NCT02039674 trial [58], Cohort F received Pembrolizumab (Pembro; 2 mg/kg) via IV infusion on Day 1 of each 3-week cycle plus Gefitinib (G; 250 mg) via oral tablets once a day every day of each 3-week cycle. Overall, seven participants were treated with this regiment, but none of these patients were able to complete the treatment due to death (*n* = 1), excluded medication (*n* = 4), or withdrawal from the study (*n* = 2).

In the NCT01472081 study [64], Arm S was treated with a combination of Nivolumab and Sunitinib, and Arm *p* was treated with a combination of Nivolumab and Pazopanib. Arm S received two different doses of Nivolumab: 2 mg/kg (SUN + NIV2; *n* = 7) and 5 mg/kg (SUN + NIV5; *n* = 26). Arm P received only 2 mg/kg dose of Nivolumab (PAZ + NIV2; *n* = 20). All-causality severe adverse effects (SAEs) of any grade were observed in 42% of the SUN + NIV2 cohort, 61% of the SUN + NIV5 cohort, and 65% of the PAZ + NIV2 cohort. Drug-related SAEs of any grade were observed in 28% of the SUN + NIV2 cohort, 46% of the SUN + NIV5 cohort, and 10% of the PAZ + NIV2 cohort. All-cause adverse effects (AEs) that led to the discontinuation of treatment were observed in 42% of the SUN + NIV2 cohort, 38% of the SUN + NIV5 cohort, and 25% of the PAZ + NIV2 cohort. These results suggested that a higher concentration of Nivolumab in combination with Sunitinib (SUN + NIV5 cohort) led to a higher chance of SAEs. By analyzing secondary responses, the SUN + NIV5 cohort achieved a lower rate of partial response to treatment, at 42%, in comparison with 71% in the SUN + NIV2 cohort. Therefore, lower concentrations of Nivolumab in combination with Sunitinib led not only to less adverse effects but also to better patient outcomes. Ideally, more trial settings are needed to study synergetic effect of two or more drugs. Unfortunately, we could not assess the synergetic effects of these treatments from the NCT01472081 study because there were no cohorts that solely underwent Nivolumab or Sunitinib treatment.

## 4. Discussion

One of the aims of this study was to find phenotypic commonalities among EGFR WT cancer patient subgroup unions that might be helpful in selecting responders for immuno-targeted combination therapies. Unlike existing “black-box” models, our approach interpreted a combination of phenotypic characteristics by assessing the frequency of significant genomic patterns in the investigated subgroups. For this purpose, we examined the third phenotypic feature for each subgroup that was a part of the unions and covered at least 15% of the dataset. For all four unions—HNSC, LUAD, LUSC, and SKCM—progression-free survival (PFS) of more than 6 months was a common feature. Interestingly, in Impower 150, clinical trial patients with EGFR WT metastatic lung adenocarcinoma had longer PFS upon combinatorial treatment with *Atezolizumab* (immunotherapy that targets PD-L1), *Bevacizumab* (targeted therapy against VEGF-A), and *Paclitaxel* and *Carboplatin* (both are chemotherapies) [66]. Finally, in the KEYNOTE-048 clinical trial, *Pembrolizumab* (immunotherapy that targets PD-L1) with chemotherapy improved overall survival versus *Cetuximab* (EGFR targeted therapy) with chemotherapy in patients with head and neck squamous cell carcinoma [67]. This demonstrated that combinatorial treatments with anti-PD-1 or anti-PD-L1 inhibitors with targeted therapy substantially prolonged the PFS of cancer patients with the EGFR WT gene.

The analysis of the dataset of Prat, A., et al. revealed that three out of six significant drug targets—FCGR2B, IGF1R, and KIT—substantially increased the odds of having stable disease versus progressive disease in EGFR WT cancer patients. The importance of this finding is further supported by the fact there is an ongoing clinical trial of *BI-1206*, a monoclonal antibody to FCGR2B, in combination with *Rituximab* (chemotherapy that targets CD20) in patients with indolent B-cell non-Hodgkin lymphoma that has relapsed or is refractory to *Rituximab* [68]. From another study, LUAD patients with high plasma levels of IGF-1 or high IGF-1R expression in tumors were associated with resistance to anti-PD-1–programmed death-ligand 1 immunotherapy, which supported the need for the clinical evaluation of IGF-1 modulators in combination with a PD-1 blockade [69]. Finally, there is an ongoing clinical trial of *Ipilimumab* (immunotherapy that targets CTLA-4) and *Imatinib Mesylate* (KIT inhibitor) for treating patients with solid tumors that have spread to other places in the body or cannot be removed using surgery [70]. Therefore, these genes could be useful therapeutic targets for immuno-targeted combination therapies.

Based on our knowledge, there is no computational pipeline that can evaluate the synergetic effect from immuno-targeted combination therapies. Our evaluations were based on primary and secondary outcomes from existing clinical trials. The limitation of these trials was that they lacked the data on monotherapy effects. For example, the NCT01472081 trial evaluated the combinations of Nivolumab + Sunitinib and Nivolumab + Pazopanib. However, there was no information on treatment with Nivolumab, Sunitinib, or Pazopanib alone. Therefore, better clinical trial design is required to objectively evaluate the synergetic effect of immuno-targeted combination therapies.

## 5. Conclusions

Phenotypic and genotypic heterogeneity are characteristic features of cancer patients that limit therapeutic response. To tackle patient heterogeneity, immuno-targeted combination therapies represent a highly promising approach for patients with no targetable mutations [71]. However, matching patient subgroups to treatment options that improve their outcome remains a challenging task. In this work, we augmented our Subgroup Discovery algorithm to identify patient subpopulations that may benefit from immuno-targeted combination therapy. Specifically, we identified drug targets that *increased* the likelihood of stable versus progressive disease in cancer patients with HNSC, LUAD, LUSC, and SKCM. Our novel informatic pipeline identified six significant drug targets and thirteen specific compounds for EGFR WT cancer patients. Three out of six drug targets—FCGR2B, IGF1R, and KIT—were previously shown to substantially increase the odds of having a stable disease in other studies. We also showed that PFS of more than 6 months was a characteristic feature among the investigated EGFR WT subgroups. Moreover, the literature demonstrated that immuno-targeted combination therapies with anti-PD-1 or anti-PD-L1 inhibitors substantially prolonged PFS in EGFR WT cancer patients. Further validation of our findings in wet-lab experiments would be a significant step toward improving healthcare for cancer patients without targetable mutations.

## Figures and Tables

**Figure 1 cancers-14-04759-f001:**
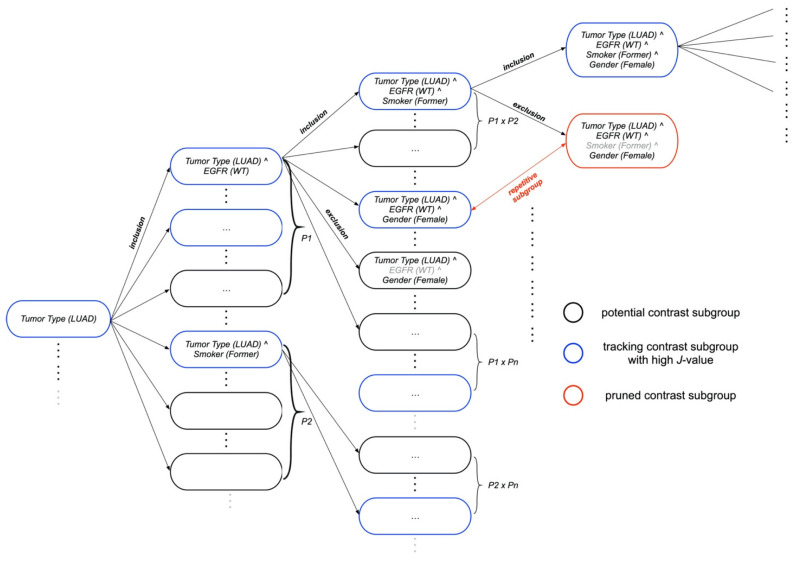
Subgroup Discovery module. The floating and path expansion processes initiate multiple, distinct starting points at various computational nodes. Based on the contrast score, the node is added or eliminated at each point. Each point portrays a potential subgroup. By applying contrast pattern mining, each candidate subgroup is scored against the outer population. *P* refers to phenotypic feature.

**Figure 2 cancers-14-04759-f002:**
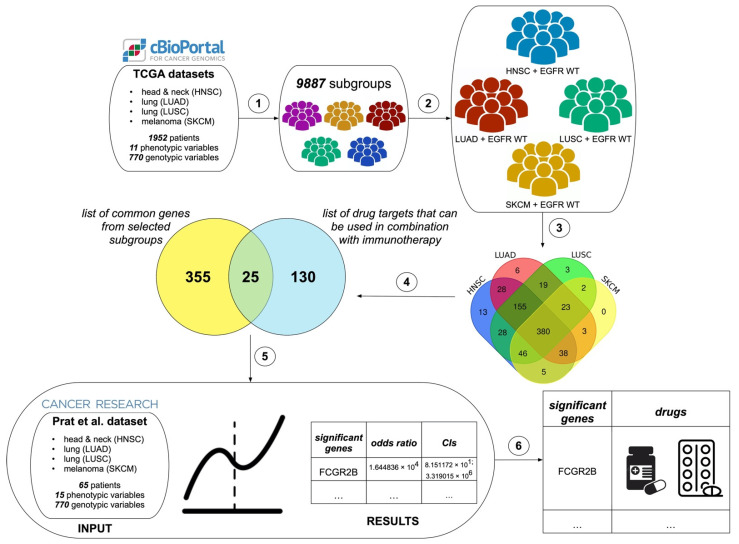
Informatic pipeline. ① We ran the Subgroup Discovery algorithm on the combined TCGA dataset. The algorithm output 9887 subgroups. ② We retained EGFR WT subgroups that covered at least 20% of the initial dataset. ③ For these subgroups, we determined common DE genes (*n* = 380). ④ We then mapped these 380 genes to the list of 155 drug targets. Thus, we identified 25 targets that could be used in immuno-targeted combination therapies. ⑤ We determined significant drug targets that increased the likelihood of SD versus PD using the proportional odds model. ⑥ We matched significant genes to drugs that could be used in immuno-targeted combination therapies using the Drug Gene Interaction Database (DGIdb).

**Table 1 cancers-14-04759-t001:** Subgroup summary for selected subgroup unions.

Subgroup Union	# Patients	% of Patients with Cancer Type	% of Patients in Whole Dataset	# Subgroups of Size Three	# Unique Genotypic Patterns	# Unique DE Genes
HNSC	500	97.08	25.66	16	8448	693
LUSC	466	96.68	23.92	16	10,005	652
LUAD	444	87.06	22.79	20	10,216	656
SKCM	407	92.29	20.89	15	7428	497

**Table 2 cancers-14-04759-t002:** Odds ratios and confidence intervals for significant genes.

	OR	2.5%	97.5%	*p*-Value
CDH5	2.875372 × 10^−2^	8.965756 × 10^−4^	9.221493 × 10^−1^	0.0448808290
FCGR2B	1.644836 × 10^4^	8.151472 × 10^1^	3.319015 × 10^6^	0.0003368373
IGF1R	2.238631 × 10^3^	1.110336 × 10^0^	4.513470 × 10^6^	0.0469308638
ITK	1.274047 × 10^−1^	2.807962 × 10^−2^	5.780693 × 10^−1^	0.0075795133
JAK2	1.047600 × 10^−5^	1.525653 × 10^−10^	7.193414 × 10^−1^	0.0435977859
KIT	3.571475 × 10^0^	1.543342 × 10^0^	8.264815 × 10^1^	0.0029426309

**Table 3 cancers-14-04759-t003:** Significant genes and potential compounds that can be used in immuno-targeted combination therapies.

Drug Target	Compound	Cancer Type
CDH5	Ruxolitinib, Lenalidomide	Lung Squamous Carcinoma, Skin Cutaneous Melanoma
FCGR2B	Bevacizumab, Cetuximab, Trastuzumab	Lung Adenocarcinoma, Head and Neck Squamous Carcinoma
IGF1R	Gefitinib	Lung Adenocarcinoma
ITK	Pazopanib, Ibrutinib	Skin Cutaneous Melanoma
JAK2	Bortezomib	Lung Adenocarcinoma
KIT	Axitinib, Cabozantinib, Pazopanib, Sunitinib	Head and Neck Squamous Carcinoma

**Table 4 cancers-14-04759-t004:** Clinical trials for predicted immuno-targeted combinations.

Trial ID	Treatment Combination	Condition	Results/Conclusions	Reference
MC1534, NCT03012230	Pembrolizumab and Ruxolitinib	Stage IV triple negative breast cancer	Estimated primary completion date: 1 April 2023.	[55]
BTCRC-HEM15-027, NCT03681561	Nivolumab and Ruxolitinib	Relapsed or refractory classical Hodgkin lymphoma	Therapy combining Ruxolitinib with Nivolumab was well tolerated and yielded encouragingly high remission rates and durable responses in patients who had all failed with previous check-point inhibitors (CPIs).	[56]
NCI-2020-08331, NCT04609046	Nivolumab and Lenalidomide	Primary CNS lymphoma	Estimated primary completion date: 31 May 2024.	[57]
MK-3475-021/KEYNOTE-021, NCT02039674	Pembrolizumab and Gefitinib	Non-small cell lung cancer	First-line Pembrolizumab plus Pemetrexed-Carboplatin continued to show improved response and survival versus chemotherapy alone in advanced NSCLC, with durable clinical benefit in patients who completed 2 years of therapy. No new safety signals were observed with longer follow-up.	[58]
MC1577, NCT03021460	Pembrolizumab and Ibrutinib	Stage III-IV melanoma	Estimated primary completion date: 1 February 2023.	[59]
OSU-18015, NCT03525925	Nivolumab and Ibrutinib	Metastatic solid tumors	Estimated primary completion date: 31 December 2021.	[60]
020-008, NCT04265872	Pembrolizumab and Bortezomib	Metastatic triple negative breast cancer	Estimated Primary completion date: 1 October 2023.	[61]
PANDORA 001, NCT04995016	Pembrolizumab and Axitinib	Locally advanced non-metastatic clear cell renal cell carcinoma	Estimated primary completion date: 20 August 2022.	[62]
Winship4234-17, NCT03468218	Pembrolizumab and Cabozantinib	Head and neck squamous cell cancer	This phase II trial of Pembrolizumab + Cabozantinib met its primary endpoint of overall response rate (ORR). The regimen was well-tolerated, with very encouraging clinical activity in relapsed metastatic HNSCC, and warranted further exploration of this disease.	[63]
CheckMate 016, NCT01472081	Nivolumab, Pazopanib, and Sunitinib	Metastatic renal cell carcinoma	The addition of standard doses of Sunitinib or Pazopanib to nivolumab resulted in a high incidence of high-grade toxicities limiting the future development of either combination regimen.	[64]
16-2300.cc, NCT03149822	Pembrolizumab and Cabozantinib	Metastatic renal cell carcinoma	This study of the combination of Pembrolizumab and Cabozantinib met the primary endpoint of ORR. Benefit was seen in first- and subsequent-line therapies. The safety profile was manageable.	[65]

## Data Availability

The TCGA dataset is available online at https://www.cbioportal.org (accessed on 9 February 2022). Prat, A., et al.’s dataset is available online at https://aacrjournals.org/cancerres/article/77/13/3540/619915/Immune-Related-Gene-Expression-Profiling-After-PD (accessed on 9 February 2022).

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
