# Peer review of "Identification of Immuno-Targeted Combination Therapies Using Explanatory Subgroup Discovery for Cancer Patients with EGFR Wild-Type Gene"

_cancers, 2022, doi:10.3390/cancers14194759_

Round 1

Reviewer 1 Report

In table 4 you should replace/add trials with safety results and add them in the script, such as:

Keynote-426, NCT02853331 

CheckMate 9ER, NCT03141177 

NCT03149822 Pembrolizumab and cabozantinib in metastatic RCC has published safety results in ASCO 2021

NCT03468218 Pembrolizumab and cabozantinib in RMHNSCC has published safety results in ASCO 2022

As you see there are a few trials with combination therapies and I don’t agree with your sentence in line 360-361.

In line 285 the numbers of the trials are the same.

I find the reference to POSEIDON trial (lines 317-321) irrelevant to your analysis, as you do not examined the combination of immune checkpoint inhibitors, but their combination with other gene-targeted drugs.

I would prefer a better reference to the results from phenotypic features of cancer patients and not only a paragraph in the discussion (lines 332-339)

Rephrase sentence ‘Immunotherapy is a type of cancer treatment that helps immune system to cure cancer’ and ‘For example, LUSC patients that are smokers with high TMB who received immune checkpoint therapy combined with chemotherapy or anti-angiogenesis therapy which had longer PFS than other participants (lines 336-339)

Author Response

  • In table 4 you should replace/add trials with safety results and add them in the script, such as:
  • Keynote-426, NCT02853331
  • CheckMate 9ER, NCT03141177

Thank you for pointing this out. We have renamed the second to last column in Table 4 from “Status of the trial” to “Results/Conclusions”. We have provided the estimated completion day for ongoing clinical trials if there were no results posted. We have also edited the clinical trials IDs in the first column in Table 4.

  • NCT03149822 Pembrolizumab and cabozantinib in metastatic RCC has published safety results in ASCO 2021

Thank you for this suggestion. We have added this clinical trial to the Table 4.

  • NCT03468218 Pembrolizumab and cabozantinib in RMHNSCC has published safety results in ASCO 2022

Thank you for your observation. We have added the updated results to the Table 4.

  • As you see there are a few trials with combination therapies and I don’t agree with your sentence in line 360-361.

Thank you for your suggestion. We have elaborated on results for most of clinical trials in Table 4.

We still believe that in NCT01472081 trial along with treatment combinations (Nivolumab + Sunitinib and Nivolumab + Pazopanib), the effect of monotherapy regimen (Nivolumab only) should be evaluated as well. This would provide a better picture about benefits of combination treatments in terms of primary and secondary outcomes.

  • In line 285 the numbers of the trials are the same.

We have corrected this mistake.

  • I find the reference to POSEIDON trial (lines 317-321) irrelevant to your analysis, as you do not examined the combination of immune checkpoint inhibitors, but their combination with other gene-targeted drugs.

Thank you for pointing it out. We removed this reference from the Discussion section.

  • I would prefer a better reference to the results from phenotypic features of cancer patients and not only a paragraph in the discussion (lines 332-339)

Thank you for your observation. We have added a paragraph to Section 3.1 regarding the significance of phenotypic features for selected subgroups.

  • Rephrase sentence ‘Immunotherapy is a type of cancer treatment that helps immune system to cure cancer’ and ‘For example, LUSC patients that are smokers with high TMB who received immune checkpoint therapy combined with chemotherapy or anti-angiogenesis therapy which had longer PFS than other participants (lines 336-339)

Thank you for your suggestion. We have edited these sentences as follows:

  • “‘Immunotherapy is a type of cancer treatment that helps immune system to cure cancer” changed to “Cancer immunotherapy is a form of cancer treatment that uses person's own immune system to prevent, control, and eliminate cancer.”
  • “For example, LUSC patients that are smokers with high TMB who received immune checkpoint therapy combined with chemotherapy or anti-angiogenesis therapy which had longer PFS than other participants” changed to “For example, LUSC patients with history of smoking and high TMB had longer PFS after treatment with chemoimmunotherapy or anti-angiogenesis therapy.”

Reviewer 2 Report

The manuscript by Dr. Kholod et al., on “Identification of immuno-targeted combination therapies using explanatory subgroup discovery for cancer patients with EGFR wild-type gene aimed to find therapeutic combinations that could extend the benefits of immunotherapy using computational method that facilitates
identification of effective combination therapies for cancer patients with few treatment options. Authors determined several specific drug targets that substantially increase the odds of stable disease versus progressive disease for head and neck cancer, lung cancer and melanoma.

The study is impressive, and manuscript is for the most part well written, the experimental progression was logical, and the data provided was comprehensive, well validated and presented clearly. Authors need to incorporate minor suggestions to improve the manuscript.

Minor concerns:

Figure legend for Figure 2 needs to be elaborated.

Please undergo a thorough check of the manuscript for typographical and grammatical errors.

Author Response

Answers for this reviewer are highlighted in blue.

The manuscript by Dr. Kholod et al., on “Identification of immuno-targeted combination therapies using explanatory subgroup discovery for cancer patients with EGFR wild-type gene” aimed to find therapeutic combinations that could extend the benefits of immunotherapy using computational method that facilitates

identification of effective combination therapies for cancer patients with few treatment options. Authors determined several specific drug targets that substantially increase the odds of stable disease versus progressive disease for head and neck cancer, lung cancer and melanoma.

The study is impressive, and manuscript is for the most part well written, the experimental progression was logical, and the data provided was comprehensive, well validated and presented clearly. Authors need to incorporate minor suggestions to improve the manuscript.

Minor concerns:

  • Figure legend for Figure 2 needs to be elaborated.

Thank you for your suggestion. We have edited the legend for Figure 2.

  • Please undergo a thorough check of the manuscript for typographical and grammatical errors.

Thank you for pointing it out. We have checked the typographical and grammatical errors.